# Construction of Reinforced Self-Cleaning and Efficient Photothermal PDMS@GDY@Cu Sponges toward Anticorrosion and Antibacterial Applications

**DOI:** 10.3390/nano13162381

**Published:** 2023-08-20

**Authors:** Yi Hu, Junmei Pu, Yingzi Hu, You Zi, Hongyan Chen, Mengke Wang, Weichun Huang

**Affiliations:** 1School of Chemistry and Chemical Engineering, Nantong University, Nantong 226019, China; 2Engineering Training Center, Nantong University, Nantong 226019, China

**Keywords:** graphdiyne, PDMS, photothermal effect, hydrophobicity, anticorrosion, antibacterial

## Abstract

Copper (Cu)-based materials are widely used in many fields from industry to life, including marine, medical apparatus and instruments, and microelectronic devices owing to their superior thermal, electrical, and mechanical properties. However, the interaction of copper with aggressive and fouling liquids under normal circumstances easily brings about severe bacterial accumulation, resulting in undesirable functionality degeneration and bacterial infections. In this contribution, we reported a novel copper-based sponge, polydimethylsiloxane (PDMS)@graphdiyne (GDY)@Cu, constructed by in situ synthesis of GDY on a commercial Cu sponge, followed by the modification of PDMS. The as-fabricated PDMS@GDY@Cu sponge not only possesses excellent self-cleaning activity against the pollution of daily drinks and dirt due to an improved static contact angle (~136°), but also display a remarkably enhanced anticorrosion performance, attributed to intimate coverage of chemically stable GDY and PDMS on the Cu sponge. Based on high photothermal effect of GDY, the PDMS@GDY@Cu sponge also displays significantly improved antibacterial activities under irradiation. In addition, due to excellent chemical stability of PDMS and GDY, self-cleaning behavior and photothermal-assisted antibacterial performance are well maintained after long-term attack of bacteria. These results demonstrate that GDY-based functional coatings hold great promises in the protection of copper devices under harsh conditions.

## 1. Introduction

Copper (Cu)-based materials are always of importance in a variety of fields ranging from industry to daily life, including marine, electronic devices and energy management due to their fascinating electrical and thermal conductivities and mechanical properties [1,2]. Unfortunately, the interaction between Cu-based materials and aggressive and fouling liquids usually brings about severe corrosion and bacterial accumulation, which finally causes a dramatic functionality decline and undesirable bacterial infections. Until now, a variety of strategies have been employed to effectively suppress copper corrosion and bacterial attachment, such as chemical vaper deposition (CVD) [3,4], organic azoles [5], polymeric coating [6], plasma [7], self-assemble monolayers [8], and electrochemical deposition [9]. Among them, polymeric coatings (e.g., epoxy [10,11], silicone [12,13], polyurethane [14], etc.), have been frequently utilized to combat the bacterial accumulation on the Cu surface owing to their high resistance to bacteria and tunable functionalities. However, these strategies are still facing some fatal challenges such as poor adhesion, inaccessible to irregular and sophisticated copper devices as well as thick films that influence the physical capabilities (e.g., electric and thermal conductivity) of the metal substrates.

Graphdiyne (GDY), an emerging carbon allotrope with an atom-thick sp- and sp^2^-hybridized all-carbon material, displays promising properties, including natural bandgap [15,16], highly conjugated and super-large π structures [17], abundant natural pores [17], high charge carrier mobility and electronic conductivity [18], superior photothermal effect [19], and excellent chemical stability [20], which hold great potential in many fields, such as electrocatalysis [21,22,23,24], solar cells [25], energy storage and conversion [26], nonlinear optics [27], etc. These outstanding physicochemical properties (e.g., antibacterial activity [28,29,30], hydrophobicity [31], and photothermal effect [32,33]) of GDY make it a promising candidate in the antibacterial applications. For example, in 2020, Zhu et al. [28] reported that GDY had a combination of “physical” and “chemical” influences on the antibacterial activity; “physical” influence played a major role in the antibacterial process because the sharp edges of GDY nanosheets (NSs) can easily insert into the bacterial membranes, resulting into the leakage of intracellular substances (e.g., K^+^ and proteins), while “chemical” influence had a trivial role due to the chemically induce reactive oxygen species production that can disturb specific bacterial processes. Moreover, in 2016, Zhang et al. [31] demonstrated that an ordered vertical honeycomb-like hydrophobic GDY foam was successfully fabricated which showed excellent oil/water separation for water remediation. In addition, in 2018, Wang et al. [33] used GDY as photo-thermo-acoustic wave nanotransducers for successfully effective photoacoustic imaging and photothermal therapy in living mice. Therefore, GDY-based nanoplatform combined with antibacterial activity, hydrophobicity, and photothermal efficiency can provide a fundamental guidance for collaboratively enhanced antibacterial performance to realize simultaneously effective protection of Cu corrosion and bacterial infections in daily life.

In this contribution, a novel copper-based sponge, polydimethylsiloxane (PDMS)@graphdiyne (GDY)@Cu, constructed by in situ synthesis of GDY on a commercial Cu sponge, followed by a facile modification of PDMS, is rationally designed and successfully fabricated. Due to excellent hydrophobicity and chemically stable structure of GDY and PDMS, the as-fabricated PDMS@GDY@Cu displays superior self-cleaning activity toward common liquids (e.g., water, cola, juice, tea and coffee), and improved anticorrosion activity in high-concentration salt solutions (e.g., 1.0 M NaCl). Combined with high photothermal effect of GDY, the PDMS@GDY@Cu presents stable photothermal performance under natural light, which largely improves the antibacterial activity compared with that in dark, suggesting the modification of GDY on the copper surface can not only remarkably suppress the oxidation but significantly boost the antibacterial behavior in daily life. Due to facile fabrication, excellent self-cleaning activity, high photothermal effect and largely enhanced anticorrosion and antibacterial activities, it is anticipated that this novel GDY-based nanoplatform can shed light on new designs of advanced nanoarchitectures for the effective protection of a variety of complicated copper devices.

## 2. Materials and Methods

### 2.1. Materials

The 1,2,3,4,5,6-hexakis[2-(trimethylsilyl) ethynyl] benzene was purchased from Shanghai Bidepharm, Co., Ltd., Shanghai, China. Pyridine (99.9%), tetrahydrofuran (THF, 99.9%) and isopropanol (IPA, 99.9%) and tetrabutylammonium fluoride (1 M in THF, 0.4 mmol) were purchased from Shanghai Macklin Biochemical Co., Ltd., Shanghai, China. Hydroxyl-terminated PDMS (Sylgard 184A) and a curing agent (Sylgard 184B) were supplied by Dow Corning Corporation, Hangzhou, China. The used Cu sponge was obtained from the market in Nantong, Jiangsu, China. *E. coli* (ATCC25922) and *S. mutans* (ATCC25175) were purchased from Shanghai Baocang Center, Shanghai, China and refrigerated at 2–8 °C and recovered at room temperature.

### 2.2. Synthesis of GDY@Cu

The Cu sponge was coated by GDY as follows: (43.6 mg, 0.066 mmol) 1,2,3,4,5,6-hexakis[2-(trimethylsilyl) ethynyl] benzene was added into 20 mL of THF. Afterwards, 0.4 mL of tetrabutylammonium fluoride solution (1 M in THF, 0.4 mmol) was added into the solution and stirred at 8 °C for 10 min. The solution was subsequently diluted with ethyl acetate, washed with brine, and dried with anhydrous Mg_2_SO_4_. The solvent was then removed under rotate evaporation, and the resulting deprotected material, hexaethynylbenzene (HEB), was re-diluted with 40 mL of pyridine. Before the reaction, the Cu sponge was treated with hydrochloric acid (10%), and then the HEB was gradually added to 50 mL of pyridine solvent containing a piece of clean Cu sponge (1 cm × 1 cm) at 60 °C, and maintained at 60 °C for 12 h or 24 h in dark, to obtain GDY@Cu-1 or GDY@Cu-2, respectively.

### 2.3. Fabrication of PDMS@GDY@Cu

An amount of 1.0 g PDMS and 0.1 g initiator were added into 10 mL of ethyl acetate. The GDY@Cu-1 or GDY@Cu-2 was immersed in the solution and subsequently taken out for curing at 80 °C for 2 h to obtain PDMS@GDY@Cu-1 or PDMS@GDY@Cu-2, respectively. For comparison, the individual modification of PDMS on the Cu sponge, abbreviated as PDMS@Cu, was also fabricated.

### 2.4. Characterization

Scanning electron microscopy (SEM, ZEISS Gemini SEM 300, Carl Zeiss AG, Oberkochen, Germany) and an EDAX system attached on the SEM were used to characterize the morphology and energy dispersive spectroscopy (EDS) of the as-obtained GDY@Cu, respectively. Transmission electron microscopy (TEM, FEI Tecnai G2 F30, Thermo Fisher Scientific, Waltham, MA, USA) was employed to characterize the morphology of GDY NSs, and high-resolution TEM (HRTEM) was utilized to characterize the atomic arrangement of GDY NSs. X-ray diffraction (XRD, D8 Advance, BRUKER AXS GMBH, Karlsruhe, Germany) signal of the as-exfoliated GDY NSs was collected using an Ultima IV X-ray diffractometer. The surface elemental composition was analyzed by X-ray photoelectron spectroscopy (XPS, Thermo Scientific KAlpha+, Thermo Fisher, Waltham, MA, USA) using an Al Kα line excitation source at an acceleration voltage of 15.0 kV. The water contact angle (CA) was characterized by an optical CA instrument (JGW-360B) using 5 µL of water droplets at room temperature, and an average value was reported for at least three tests. The anticorrosion activity was characterized by electrochemical impedance spectroscopy (EIS) in the frequency range of 1 to 10^5^ Hz at an amplitude of 0.005 V. The photothermal performance was evaluated under irradiation by a xenon short arc lamp with a wavelength range from 800 to 1100 nm (Appendix A), and a thermal imaging camera (ST 9450A+, Smart Sensor) was used to take the thermal photographs. *S. mutans* (Gram-positive bacteria) and *E. coli* (Gram-negative bacteria) were selected in the antibacterial experiment. The as-prepared samples were disinfected in 75% (*v*/*v*) ethanol/water solution for 30 min, and subsequently placed in a 12-well plate and irradiated under a UV lamp for 1 h. Afterwards, 1 mL bacteria suspension with a concentration of 1 × 10^8^ CFU mL^−1^ was added to the corresponding samples. After incubating at 37 °C for 3 h, all the masks were repeatedly washed with sterilized water 3 times to remove the loosely adhered bacteria. The washed samples were then immersed in 4 mL sterilized water and treated by ultrasonication for 1 min to peel off the bacteria on the surface of the as-prepared samples. An amount of 100 μL ultrasonic solution was dropped on the LB agar plate and incubated at 37 °C for 24 h, and the colony number was counted via a standard surface plating method. To test the photothermal sterilization performance of the as-prepared samples, the samples, after being washed with sterilized water 3 times, were irradiated under 0.1 W cm^−2^ for 60 s, and the following treatments were the same as the aforementioned steps. For the observation of bacteria morphologies, the samples after being washed with sterilized water 3 times in the antibacterial experiments were immobilized by 2.5% glutaraldehyde solution overnight. After drying, the bacteria morphologies were observed by SEM measurement.

## 3. Results and Discussion

Figure 1a presents the schematic diagram of the synthesis of the PDMS@GDY@Cu, including two steps: (i) in situ synthesis of GDY on the Cu sponge using HEB as a monomer at 60 °C; (ii) followed by the PDMS modification. The photographs of the pristine Cu sponge and modified samples can be seen in Figure 1b. Only the PDMS@Cu still exhibits the natural color of Cu sponge, while others present remarkably darker color, suggesting the successful cover of GDY on the Cu surface. SEM image (Figure 1c) of the as-fabricated GDY@Cu-2 displays a similar morphology to that on the Cu nanowires [34], which is much rougher than that of pristine Cu (Appendix A). EDS analysis in Appendix A of the GDY@Cu-2 found that there are three elements (C, Cu and O). Note that a tiny amount of O could be due to the oxidation of some terminal alkynes [35]. For better understanding of the surface modification, the as-fabricated GDY@Cu-2 stored in IPA was gently sonicated in an ice bath for 30 min, and the obtained suspension was centrifugated to obtain the precipitate. The TEM image of the precipitate shows two very thin nanosheets with a lateral size ranging from ~250 nm to ~420 nm (Figure 1d). HRTEM image (Figure 1d inset) shows a clear fringe lattice of 0.45 nm, in good agreement with the previously reported GDY [36]. Furthermore, XRD pattern of the precipitate shows a main peak at ~23.4° (Figure 1e), which could be assigned to the C (002) [37], suggesting the multiple packing mode of GDY. XPS was carried out to investigate the structure and elementary composition of the as-obtained precipitate, shown in Figure 1f,g. The XPS survey in Figure 1f shows predominant elemental carbon and a tiny amount oxygen, in good accordance with the EDS result (Appendix A). The high-resolution asymmetric C 1s XPS spectrum in Figure 1g can deconvoluted into four Gaussian curves with major contributions at 284.6 eV and 285.1 eV, which could be indexed to the orbitals in C–C (sp^2^) and C–C (sp), respectively, while minor contributions at 286.4 eV and 288.1 eV can be assigned to C–O and C=O species, respectively. The areal ratio of the C–C (sp)/C–C (sp^2^) is close to 2.0, which matches well with the chemical composition of GDY. It should be pointed out that the calculated ratio (~0.21) of O/C is similar to those of GDY synthesized by other approaches (~0.2) [35,38], and other carbon nanomaterials, such as CVD-grown graphene (~0.2) [39]. All the structure characteristics confirm the successful synthesis of GDY NSs on the surface of Cu sponges.

The CA was carried out to characterize the wetting property of the GDY@Cu, as shown in Figure 2a,b. It can be observed that the wettability changes from hydrophobicity (~127.4°) to hydrophilicity (0°) after the modification of GDY due to the small amount of O introduced in the synthetic process and hierarchical nanostructures of GDY. The hydrophilicity after the GDY modification may arise from the introduction of polar functional groups such as oxygen atoms or hydroxyl groups due to the oxidation of terminal alkynes. These functional groups are capable of forming hydrogen bonds with water molecules, and thereby enhancing its hydrophilicity [40]. Previous work demonstrated that PDMS could remarkably reduce the surface energy of substrates for efficient self-cleaning, and simultaneously enhance their chemical and mechanical stability [41,42]. Here, after the individual incorporation of PDMS, the hydrophobicity of the PDMS@Cu (136.4°, Appendix A) slightly improves compared to that for the pristine Cu sponge (127.4°). Note that the hydrophobicity of both the PDMS@GDY@Cu-1 and PDMS@GDY@Cu-2 also improves compared to that of pristine Cu sponge, same as the PDMS@Cu and the reported results [43], indicating the boosted self-cleaning performance. Moreover, to evaluate the anticorrosion activity, EIS technique was employed to study the influence of GDY and PDMS coverage on the corrosion activity in high-concentration salt solutions (e.g., 1.0 M NaCl), as shown in Figure 2c. It can be observed that both of the GDY@Cu-1 and PDMS@GDY@Cu-1 show more stable interfacial resistance (*R*) compared with the pristine Cu sponge before and after 7-day corrosion, i.e., the *R*s for the fresh samples of GDY@Cu-1 and PDMS@GDY@Cu-1 are 8.2 Ω and 34.8 Ω, respectively, while increase to 9.4 Ω and 36.1 Ω, respectively, after 7-day corrosion; both the changes of *R* (Δ*R*s) for the GDY@Cu-1 (Δ*R* = 1.2 Ω) and PDMS@GDY@Cu-1 (Δ*R* = 1.3 Ω) before and after 7-day corrosion are significantly lower than that for the pristine Cu sponge (Δ*R* = 6.3 Ω) under the same conditions. The photographs in Appendix A indicate that the colors of the GDY@Cu-1 and PDMS@GDY@Cu-1 are nearly similar before and after 7-day corrosion while the color of the pristine Cu sponge becomes darker, confirming that only the pristine Cu sponge have a severe corrosion in 1.0 M NaCl. Both the EIS and digital photograph result confirm that the coverage of chemically stable GDY and PDMS have a great contribution to the anticorrosion activity of the Cu-based devices.

The improved hydrophobicity endows PDMS@GDY@Cu with promising properties in self-cleaning against daily pollutants. Figure 3a shows the images of a variety of drink droplets setting on the PDMS@GDY@Cu-2, including water, cola, juice, tea, and coffee. All the selected drinks display spherical shapes on the PDMS@GDY@Cu-2. In addition, the surface of PDMS@MOF@Cu-2 can stay dry after the sample that were stuck on a clean glass immediately taken out from the aforementioned drinks (Figure 3b). Due to the low surface energy, hydrophobic PDMS@GDY@Cu-2 exhibits excellent self-cleaning capability when the surface is severely contaminated by salts (Figure 3c) and dirt (Figure 3d). In both cases, the pollutants such as salts and dirt could be readily cleaned by a small quantity of water (Appendix A), which holds great potentials in the antibacterial community and water economy.

Due to excellent photothermal effect of GDY [32,33], the photothermal performance of the as-fabricated GDY-based samples including GDY@Cu-1, GDY@Cu-2, PDMS@GDY@Cu-1 and PDMS@GDY@Cu-2 were investigated. For clarity, the photothermal performance of the pristine Cu sponge and PDMS@Cu sample was also studied. It can be observed in Figure 4a that the surface temperatures for all the samples gradually increases within 5-min light illumination, and then they reach an equilibrium state. Compared with the pristine Cu sponge, the PDMS@Cu shows a larger surface temperature, mainly attributed to the thermal insulation of PDMS. More importantly, the incorporation of GDY into the Cu sponges dramatically enhances the photothermal conversion, i.e., as shown in Figure 4b, after 5-min light illumination, the equilibrium surface temperatures for the GDY@Cu-1, GDY@Cu-2, PDMS@GDY@Cu-1 and PDMS@GDY@Cu-2 are 82.0 °C, 91.8 °C, 86.3 °C and 95.5 °C, respectively, which are significantly higher than those for the pristine Cu sponge (40.1 °C) and PDMS@Cu (56.5 °C). The influence of the GDY content on the surface temperature of GDY@Cu demonstrate that the photothermal performance enhances with the GDY content, and the trend maintains after the modification of PDMS, indicating that the photothermal performance of the GDY-modified Cu-based devices can be easily modulated by simply tuning the GDY content. Here it should be noted that under the same conditions, the fact that the surface temperatures of the PDMS@GDY@Cu samples (PDMS@GDY@Cu-1 and PDMS@GDY@Cu-2 are apparently higher than the corresponding precursors ((GDY@Cu-1 and GDY@Cu-2), is similar to the above-mentioned phenomenon caused by the Cu sponge and PDMS@Cu. In addition, the on/off illumination cycling for all the studied samples in Figure 4c shows that the as-fabricated GDY@Cu and PDMS@GDY@Cu present very stable photothermal conversion. The superior photothermal performance of GDY can be ascribed to the narrow direct band gap energy (0.46 eV) [44] in association with nonradiative relaxation. The 2D GDY NSs on the Cu sponge can easily absorb photons, and excite the electron transition, and subsequently the electrons return to the low-energy states with the energy released via radiating or nonradiative phonons. In the nonradiative process, the heat is generated as the phonons hit the surface lattice of GDY, leading to a temperature gradient.

Although the GDY has shown antibacterial activity from the aspects of “physical” property by physically destroying the bacterial membrane [28], and “chemical” property by efficiently transfer electrons to bacteria, consequently facilitating the generation of reactive oxygen species [28], yet the study on antibacterial activity of GDY-based nanostructures is still at its initial stage. It can be seen in Figure 5a–d that after the PDMS coated on the Cu sponge, the accumulated bacteria number largely decreases compared to that for pristine Cu sponge because of the significantly enhanced hydrophobicity. Moreover, the individual modification of GDY into the Cu sponges (e.g., GDY@Cu-1 and GDY@Cu-2) has a negative effect on the antibacterial activity towards both *E. coli* and *S. mutans* (Figure 5a,c), mainly due to their excellent hydrophilicity, confirmed by the bacteria number after being treated by GDY@Cu-1 and GDY@Cu-2. In both cases of *E. coli* and *S. mutans*, it shows that the hydrophobicity makes a great contribution to the antibacterial activity compared with the GDY component.

Surprisingly, with the marriage of PDMS and GDY, the PDMS@GDY@Cu shows superior antibacterial activity to GDY@Cu and PDMS@Cu due to the collaborative effect of antibacterial activity from GDY and boosted hydrophobicity from PDMS. Upon light illumination, all the GDY-based Cu sponges display outstanding antibacterial activity towards both *E. coli* and *S. mutans* compared with those in dark (Figure 5a–d), which can be attributed to high photothermal conversion of GDY. In addition, it can also be observed that the antibacterial activity is further enhanced on light illumination with the increase in GDY content because of improved photothermal effect, which is comparable to thermal induced antibacterial result at ~60 °C in both cases of *E. coli* (Appendix A) and *S. mutans* (Appendix A), demonstrating the effective photothermal-assisted antibacterial performance based on GDY that could boost the generation of reactive oxygen species (e.g., •OH, •O_2_^−^, ^1^O_2_, and H_2_O_2_) within bacterial cells to induce damage to both DNA and proteins [45,46]. Note that the physical antibacterial activity for GDY itself in the sample PDMS@GDY@Cu is not satisfied in this project due to the outer modification of GDY by PDMS.

The stable antibacterial behavior is also of great importance for long-term use of Cu-based devices in practical applications. Figure 6a presents the photographs of the bacteria, including *E. coli* and *S. mutans*, treated by the PDMS@GDY@Cu-1 for two weeks with and without light illumination. It can be seen in Figure 6b that the PDMS@GDY@Cu-1 exhibits similar antibacterial result to the fresh sample in Figure 5b,d, evidenced by the flat thin NS morphology collected by the sonication of PDMS@GDY@Cu-1 after two-week antibacterial experiment (Appendix A), verifying high stability of the PDMS@GDY@Cu-1. This, combined with improved hydrophobicity and photothermal-assisted effect, indicates that the PDMS@GDY@Cu holds great promises in Cu-based biomedical applications such as practical biomaterial scaffold and biological packaging materials.

## 4. Conclusions

In this study, a novel copper-based sponge, PDMS@ GDY@Cu, constructed by in situ synthesis of GDY on a commercial Cu sponge, followed by the modification of PDMS. The GDY that grew on the Cu sponge has a lateral size ranging from ~250 nm to ~420 nm. The wettability changes from hydrophobicity (Cu sponge) to hydrophilicity (GDY@Cu) to improved hydrophobicity (PDMS@GDY@Cu). Both the GDY@Cu and PDMS@GDY@Cu samples exhibit superior anticorrosion activity in 1.0 M NaCl to the pristine Cu, suggesting that the intimate coverage of chemically stable GDY and PDMS on the Cu sponge can indeed largely improve the lifetime of the Cu devices. The as-prepared PDMS@GDY@Cu not only displays excellent self-cleaning behavior but exhibits stable photothermal performance with a maximum surface temperature of 95.5 °C. Based on highly efficient photothermal effect of GDY and improved hydrophobicity, the PDMS@GDY@Cu sponge displays outstanding antibacterial activities. In addition, the self-cleaning behavior and photothermal-assisted antibacterial performance are well maintained after long-term attack of bacteria such as *S. mutans* and *E. coli*. It is envisioned that this research can pave the way to elaborate hierarchical structures on Cu solid substrate by a variety of approaches for the protection of copper devices under harsh conditions.

## Figures and Tables

**Figure 1 nanomaterials-13-02381-f001:**
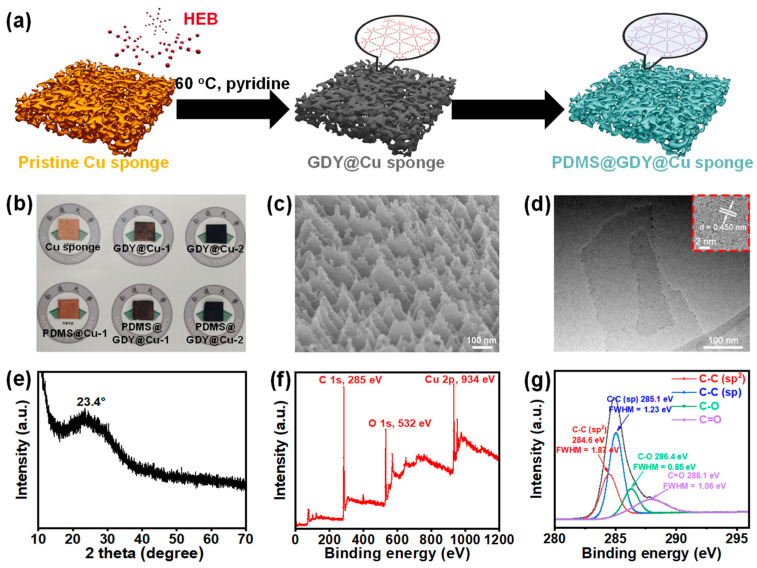
Schematic diagram and structural characterization of the as-fabricated PDMS@GDY@Cu sponges. (**a**) Schematic diagram of the PDMS@GDY@Cu sponges; (**b**) photographs of the pristine Cu sponge and modified Cu sponges; (**c**) SEM image of the GDY@Cu-2; (**d**) TEM image and (**e**) XRD pattern of the GDY obtained by the exfoliation of GDY@Cu-2; inset showing its HRTEM image; (**f**) XPS spectrum of the GDY@Cu-2 and (**g**) its narrow scan for the C 1s orbital.

**Figure 2 nanomaterials-13-02381-f002:**
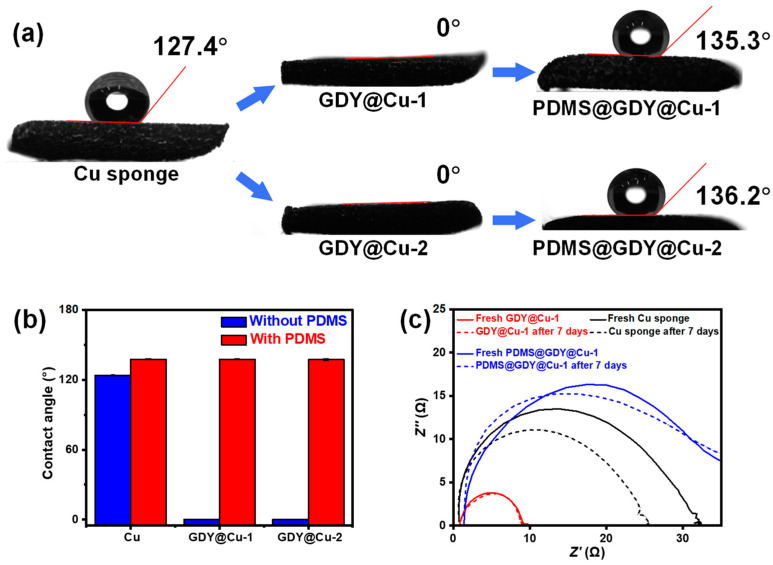
(**a**) Wettability of the studied samples and (**b**) the water CA changes before and after the modification of PDMS. (**c**) EIS spectra of the studied samples in 0.1 M NaCl.

**Figure 3 nanomaterials-13-02381-f003:**
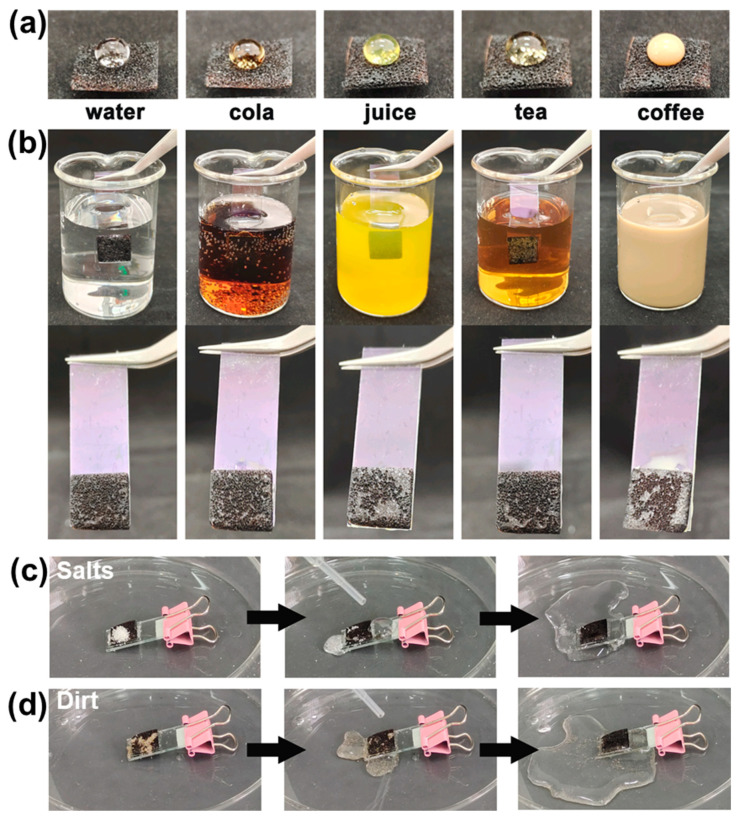
(**a**) Optical images of a variety of common liquid droplets setting on the as-fabricated PDMS@GDY@Cu sponges; (**b**) optical images of PDMS@GDY@Cu sponges after taken out from diverse drinks. Self-cleaning process on the polluted surface of PDMS@GDY@Cu mesh by (**c**) salts and (**d**) dirt.

**Figure 4 nanomaterials-13-02381-f004:**
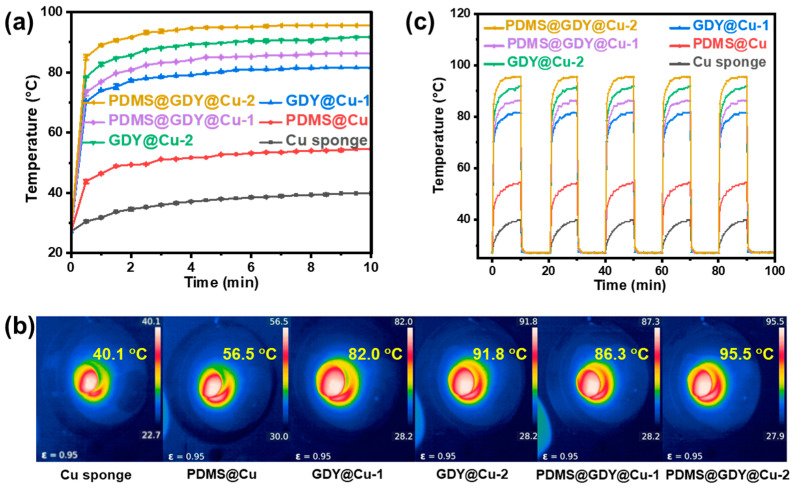
(**a**) Surface temperature variations of the studied samples irradiation at 250 mW cm^−2^ for 10 min and (**b**) surface infrared thermal pictures of the studied samples. (**c**) their corresponding surface temperature variations for five light on/off consecutive cycles.

**Figure 5 nanomaterials-13-02381-f005:**
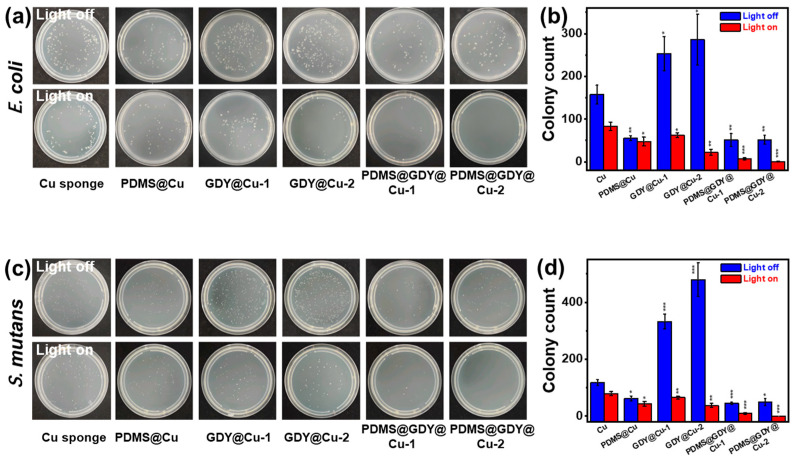
Photographs of the (**a**) *E. coli* bacterium treated by the studied samples with and without light illumination, and (**b**) the corresponding data analysis. Photographs of the (**c**) *S. mutans* bacterium treated by the studied samples with and without light illumination, and (**d**) the corresponding data analysis. For better quality, the *E. coli* and *S. mutans* bacteria were illuminated at 145 mW cm^−2^ for 15 min and 88.0 mW cm^−2^ for 8 min, respectively (all experimental results were *p*-tested, * *p* < 0.05, ** *p* < 0.01, *** *p* < 0.001).

**Figure 6 nanomaterials-13-02381-f006:**
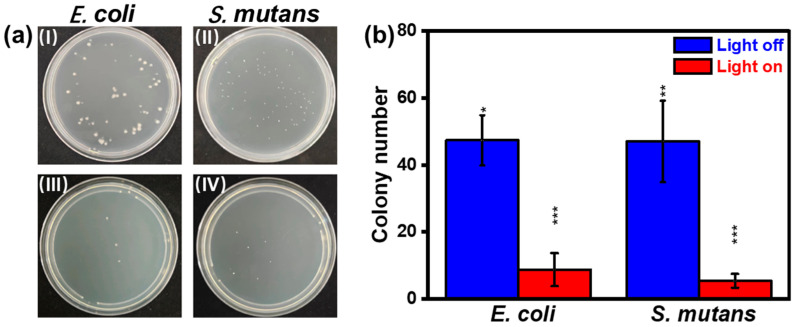
(**a**) Photographs of *E. coli* and *S. mutans* bacteria treated by the PDMS@GDY@Cu-1 after two weeks without (**I**,**II**) and with (**III**,**IV**) light illumination, and (**b**) their corresponding data analysis of *E. coli* and *S. mutans* bacteria. For better quality, the *E. coli* and *S. mutans* bacteria were illuminated at 145 mW cm^−2^ for 15 min and 88.0 mW cm^−2^ for 8 min, respectively (all experimental results were *p*-tested, * *p* < 0.05, ** *p* < 0.01, *** *p* < 0.001).

## Data Availability

Not applicable.

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
