# Peer review of "Construction of Reinforced Self-Cleaning and Efficient Photothermal PDMS@GDY@Cu Sponges toward Anticorrosion and Antibacterial Applications"

_nanomaterials, 2023, doi:10.3390/nano13162381_

Round 1

Reviewer 1 Report

Manuscript ID: nanomaterials-2550362
Type of manuscript: Article
Title: Construction of Reinforced Self-Cleaning and Efficient Photothermal
PDMS@GDY@Cu Sponges toward Anticorrosion and Antibacterial Applications
Authors: Yi Hu, Junmei Pu, Yingzi Hu, You Zi, Hongyan Chen, Mengke Wang *,
Weichun Huang

The manuscript reports the synthesis and investigations of PDMS on graphydine on Cu sponges, showing photo-thermal, anticorrosion and antibacterial properties.

I find the work interesting and well written that deserves publication in Nanomaterials after review, following the comments and suggestions listed hereafter, containing somethings formal and others more essential.

In my opinion, the title could be changed to a more general one, where instead of PDMS@GDY@Cu, one could write functionalized-Cu sponges with graphdyne and PDMS,and the word construction changed to: Synthesis, for instance. I feel that the PDMS@GDY@Cu is too cryptic.

Add in the key words: PDMS.

At page 4 Report the text before the Figure 1, as well as for Figure 2.

Add  details of the fit procedure  for XPS fit of C core level (Fig.1 g) (FWHM of Gaussian and Lorentzian of the Voigt functions).

In figure 2 indicate the wetting angle.

Try to avoid Fig1A&Fig.1b replacing & with: and  (there are many disseminated within the text).

Modify the caption of Figure 5. It is a bit unclear.

Reinforce the discussion on the antibacterial properties for GDY@Cu at page 8, about Figure 5, giving the physical reasons of this behaviour (also for PDMA@GDY#Cu sample).

Give, in the text, details about illumination used.

In my opinion, the work will be accepted for publication in Nanomaterials from MDPI, after a minor revision based on the suggestions, comments here proposed.

Minor English revision is requested (some grammar errors are present).

Reviewer 2 Report

In this study, the authors synthesized GDY on Cu sponge and subsequently modified it with PDMS. The adhesion of chemically stable GDY and PDMS on Cu sponges significantly improved the lifetime of Cu devices and showed excellent antimicrobial activity. This research paper reports some interesting findings and is recommended for publication in Nanomaterials.

Some considerations are listed below.

1) I am not sure what you mean by "natural bandgap". Please explain in more detail.

2) I have a question regarding the following statement. 

 EDS analysis in Figure S3 of the GDY@Cu-2 found that there are three elements (C, Cu and O). Note that a tiny amount of O could be due to the oxidation of some terminal alkynes.

What about the possibility that the Cu surface is oxidized? 

3) Is there a molecular-level explanation for the increased hydrophilicity by GDY? Is it caused by hydrogen bonding? Or is it caused by the dispersion force? That is what is discussed in the following paper.

https://doi.org/10.1021/acs.langmuir.1c01935
